# Online and Offline Behavior Change Techniques to Promote a Healthy Lifestyle: A Qualitative Study

**DOI:** 10.3390/ijerph19010521

**Published:** 2022-01-04

**Authors:** Daniël Bossen, Monique Bak, Katja Braam, Manon Wentink, Jasmijn Holla, Bart Visser, Joan Dallinga

**Affiliations:** 1Centre of Expertise Urban Vitality, Faculty of Health, Amsterdam University of Applied Sciences, 1105 BD Amsterdam, The Netherlands; m.a.bak@hva.nl (M.B.); Katja.braam@inholland.nl (K.B.); manonwentink@hotmail.com (M.W.); b.visser2@hva.nl (B.V.); j.m.dallinga@hva.nl (J.D.); 2Faculty of Health, Sports and Social Work, Inholland University of Applied Sciences, 2015 CE Haarlem, The Netherlands; jasmijn.holla@inholland.nl; 3Amsterdam Rehabilitation Research Center, Reade, 1054 HW Amsterdam, The Netherlands

**Keywords:** combined lifestyle interventions, overweight, behavior change techniques, eHealth

## Abstract

Combined lifestyle interventions (CLI) are focused on guiding clients with weight-related health risks into a healthy lifestyle. CLIs are most often delivered through face-to-face sessions with limited use of eHealth technologies. To integrate eHealth into existing CLIs, it is important to identify how behavior change techniques are being used by health professionals in the online and offline treatment of overweight clients. Therefore, we conducted online semi-structured interviews with providers of online and offline lifestyle interventions. Data were analyzed using an inductive thematic approach. Thirty-eight professionals with (*n* = 23) and without (*n* = 15) eHealth experience were interviewed. Professionals indicate that goal setting and action planning, providing feedback and monitoring, facilitating social support, and shaping knowledge are of high value to improve physical activity and eating behaviors. These findings suggest that it may be beneficial to use monitoring devices combined with video consultations to provide just-in-time feedback based on the client’s actual performance. In addition, it can be useful to incorporate specific social support functions allowing CLI clients to interact with each other. Lastly, our results indicate that online modules can be used to enhance knowledge about health consequences of unhealthy behavior in clients with weight-related health risks.

## 1. Introduction

In the Netherlands, more than 50% of the adult population is currently moderately or severely overweight [1]. This prevalence rate is in line with other high- and middle-income countries [2]. Overweight is associated with a range of health issues and noncommunicable diseases, including type-2 diabetes, cardiovascular disease, cancer, and physical and psychological impairment [3]. Although underlying determinants are complex and multifactorial, overweight is a result of an imbalance in energy balance-related behaviors, namely physical activity and food intake [4].

Combined lifestyle interventions (CLI) are focused on guiding clients with weight-related health risks into a healthy lifestyle. Research indicates that CLIs support individuals in initiating and maintaining a healthy lifestyle [5,6,7]. The multi-component approach, targeting eating and physical activity behaviors, contributes to improvements in cardiovascular and metabolic factors, including weight, HbA1c, waist circumference, and cholesterol [8,9].

In 2019, three effective CLIs were included by the basic health insurance package in the Netherlands [10,11,12]. These 2-year interventions contain a behavioral change (year 1) and a maintenance phase (year 2) and involve individual and group coaching sessions by a health professional, physical therapist, and/or dietician to achieve long-term lifestyle changes. CLI interventions use different behavior change techniques (BCTs) to regulate a target behavior [13]. BCTs are components of behavior change interventions that can independently bring change in a person’s behavior under favorable circumstances. BCTs may alter or redirect causal processes that regulate physical activity and nutrition behavior. They describe mechanisms with actions to support behavior change [14]. Although CLIs incorporate specific techniques intended to alter the behavior of clients with weight-related health risks, it is currently unclear how BCTs are used by professionals in CLIs. This limits interpretation and replication for future interventions and the understanding of how BCTs relate to effectiveness [15].

In addition to the moderating role of BCTs on the effectiveness of CLIs, the mode of delivery may also play a crucial role. CLIs are currently mainly delivered in face-to-face sessions without or with limited use of eHealth. This is a pity because eHealth, especially in combination with face-to-face, is promising in long-term interventions to guide clients toward a sustainable healthy lifestyle [16,17,18]. eHealth refers to health services and information delivered or enhanced through internet-related technologies [19]. eHealth interventions permit digital monitoring of important health parameters including physical activity and calorie intake either through self-report or by objective performance data. Based on these data, eHealth interventions can include various online or digital behavior change support tools to nudge clients in their decision-making processes, such as goal setting in a web-based or mobile application with respect to a desired behavior and outcome (e.g., make an action plan), social support through online groups (e.g., communicate with important others), and monitoring (e.g., pedometer on a phone or smartwatch, reporting weight changes in a web-based or mobile monitoring tool with or without feedback from a coach). Moreover, data about exercise and food intake provide important insight in clients’ progress for a tailored coaching trajectory, beyond the walls of the practice. However, although promising, many eHealth studies report limited or inconsistent results. These disappointing findings are often related to poor adherence rates and a lack of clarity which mechanisms and BCT components of eHealth interventions support behavior change [20].

In response to the current COVID-19 pandemic, professionals shift rapidly from physical to online coaching and use more eHealth without knowledge of which online BCTs are most suitable [21]. In order to integrate eHealth into existing CLIs, it is important to identify which BCT’s are currently being used by health professionals in the treatment of overweight clients and how these techniques may be applicable in coaching trajectories. Therefore, this study aims to answer the research question: “How are online and offline BCTs used by health professionals in CLIs to improve physical activity and eating behaviors in clients with weight-related health risks?”.

## 2. Materials and Methods

### 2.1. Design

To answer the research question, this study used a qualitative research design consisting of semi-structured interviews. Positive advice on the research protocol was provided by the Ethical Committee of the Amsterdam University of Applied Sciences (project number 200718) on 18 July 2020. The Consolidated Criteria for Reporting Qualitative Research checklist was used to secure accurate and complete reporting [22] (Appendix A). This study is part of the iTLC project that aims to develop a method that can be used to assist professionals in using eHealth in existing CLIs.

### 2.2. Participants

Professionals were eligible for inclusion if they worked as a health professional or developed a lifestyle intervention for clients with an increased health related risk. These participants were recruited in two rounds. In the first round, we recruited certified professionals that delivered a CLI covered by the Dutch basic health insurance. These professionals had a background in physical therapy, exercise therapy, lifestyle coaching, or nutrition and had no or minimum experience with the use of eHealth. The recruitment in this round took place through the network of the Dutch Dietetic Association, the Professional Association of Lifestyle Coaches, and two organizations that deliver lifestyle interventions. In the second round, we recruited professionals that use eHealth interventions as a mode of communication to get information about clients’ behavior or give feedback. These professionals were recruited through two online lifestyle companies and the social network of the authors. Recruitment was concurrent with data collection and continued until saturation was achieved. Researchers had no or a minimum relationship with the participants prior to study commencement. All interested participants were provided with an information sheet, in which goal, planning, content of the study, the target group, data management, and privacy aspects were explained, and an informed consent form was provided. Once written informed consent was obtained, participants were enrolled in the study.

### 2.3. Data Collection

Directly before study commencement, COVID-19 restrictions were implemented across the Netherlands. This meant that face-to-face interviews were organized online instead of in person. The online interviews were conducted between October 2020 and February 2021, using Microsoft Teams (Microsoft Corporation, Albuquerque, NM, USA). The semi-structured interviews were conducted by the first (male, researcher, PhD), second (female, researcher, MSc), third (female, researcher, PhD), and fourth (female, researcher, PhD) author. These researchers are experienced in conducting qualitative research. The interviews followed a semi-structured topic guide developed in collaboration with seven professionals (Appendix A). This guide was based on existing literature and related theories [13,14,23] and consisted of the following main areas: (1) use of BCTs in practice, (2) ways of using behavior change techniques in online and offline coaching, and (3) requirements for successful use of online and offline behavior change techniques. The first part of the interview guide (the use of BCTs in practice) was based on the taxonomy of BCTs by Michie et al. [14]. At the start of the interview, each participant received a summary of the study background. Subsequently, sociodemographic data (age, gender, profession) were collected, and participants were asked a number of general questions about their work activities and their experience with the target group and eHealth. Thereafter, questions about the main areas were posed, albeit in varying order. Interviews lasted approximately 60 min, were video and audio recorded, and were transcribed verbatim by a research assistant who was not involved in the study. All data were stored in digital format on a secured university server.

### 2.4. Data Analysis

The data were analyzed through an inductive thematic approach based on the six-step process by Braun and Clarke [24]. Data were analyzed by the same researchers (DB, MB, KB, and MW) who conducted the interviews. First, the transcribed interviews were individually checked (against the recorded interviews) and read by the researchers to familiarize themselves with the data (step 1). Then, segments of the transcripts were identified and classified as open codes (step 2). Recurrent patterns were identified and sorted into preliminary (sub)themes (step 3). These identified themes were reviewed by two pairs of researchers (DB&MW and KB&MB) and refined afterwards (step 4). We subsequently presented the themes in several discussion meetings with all authors, which resulted in the definitive structure and names of the (sub)themes (step 5, Appendix A). The ultimate themes are presented in this manuscript (step 6). Prior to the analysis, the above-described steps were recorded in a protocol. MAXQDA-Analytics Pro 2020 (VERBI Software, Berlin, Germany) was used to analyze the data.

## 3. Results

### 3.1. Participant Characteristics

A total of 38 participants were interviewed, 20 males and 18 females (Table 1). The average age of participants is 38.6 years (SD 10.6). Most participants are lifestyle coaches (*n* = 14) or exercise therapists (*n* = 10). The sample consists of professionals who have experience with the (1) provision of the traditional ‘offline’ CLIs (*n* = 14), (2) provision of online CLIs (*n* = 14), and (3) the development of (online) CLIs (*n* = 10).

### 3.2. Behavior Change Techniques

Data analyses showed various used BCTs. BCTs were used in both offline and online sessions to help and encourage behavior change in clients with an increased health-related risk. We observed differences in how BCTs are used in practice between the groups of professionals. CLI providers, both online and offline coaches, are predominately practically oriented. The offline coaches use evaluative questions to tailor their treatment, while online coaches use specific tools to obtain objective performance measures for feedback, monitoring, or goal-setting purposes. Developers of (online) CLIs are more theoretically oriented and aware of how they use specific BCTs in practice, while the online and offline coaches use BCTs in a more unconscious way. This also required a more in-depth inquiry during the interviews with illustrative examples to address specific behavior change techniques.

The ultimate themes are labeled according the taxonomy of behavior change techniques by Michie et al. (2013) [14]. Four main themes were identified by the analysis: (1) goal setting and action planning, (2) providing feedback and monitoring, (3) facilitating social support, and (4) shaping knowledge. The findings for each theme describe how online and offline BCTs are used by health professionals to improve physical activity and eating behaviors in clients with weight-related health risks. We will address each of these themes, in turn, from the health professional perspective.

### 3.3. Goal Setting and Action Planning

Professionals indicate that goal setting is an important BCT strategy to promote healthy eating and physical activity behavior. According to the professionals, setting a goal in terms of the behavior achieved and making a detailed action plan are most promising. Setting realistic and personally relevant goals is applied within both online (e.g., an online mobile application or a CLI dashboard) and/or offline (e.g., paper diary) tools. Goals are preferably formulated by clients themselves. They need to offer a realistic prospective for the future and prevent the under- or overestimation of abilities. Professionals argued that a realistic action plan is important to boost clients’ confidence. As one professional stated: “*So, we use very small steps to prevent clients in going too fast. We want to keep it very realistic and sustainable; people like to create big goals for themselves, to challenge themselves. That results in failure, while if you set a goal for one week and you reached it at the end of that week, you can be very proud on yourself*” (P#24, female). Professionals also indicate that goal setting is a crucial element in online behavior change interventions, allowing users to set, monitor, or review goals related to a target behavior. *“Together, we set a long and short goal which are visible in the online application. Based on the short-term goal, weekly modules are generated. This provides guidance which make clients more determined to engage in physical activity”* (p#23, female).

### 3.4. Feedback and Monitoring

According to the professionals interviewed, the provision of feedback is emphasized as a motivator for behavior change. Professionals monitor and provide informative or evaluate feedback on (the outcome of) performance of the behavior. Monitoring was predominantly used for data collection purposes as input for feedback strategies rather than a strategy aimed at changing behavior. Feedback is used to provide clients with data about their lifestyle, instructions how to reach improvement in health behavior, or reflections on a person’s behavioral performance. Feedback with suggestions of strategies to improve success and prevent relapse are included in the existing lifestyle interventions. Video consultation is seen as a convenient route to provide just-in-time feedback at critical moments during the treatment. “*I am a lot more flexible since I use online consultations, it gives me more possibilities to interact with my clients*” (P#3, female). Professionals indicate that feedback should be personalized, goal oriented, and based on actual performed behavior. However, the feedback provided by health professionals as part of the lifestyle interventions is mostly based on clients’ stories rather than objective performance data. As one professional put it: “*I don’t monitor exercises, fat intake or other health indicators. I do not have the sources to track these things*” (P#4, female). Professionals who do use online behavior change interventions point out that activity trackers enable them to provide accurate feedback. Professionals use different applications to monitor food intake (e.g., calory tracker) or physical activity (e.g., fitbit, MyFitnessPal, and an application named ‘Ommetje’ (Dutch for ‘Detour’)). These applications are particularly promising for collecting objective and continuous physical activity parameters (e.g., steps, minutes, intensity) and have the ability to transfer data to online platforms and mobile phones. Professionals indicate that they prefer objective data monitoring systems over subjective measures (e.g., questionnaires or diaries). Data that are generated through pedometers or accelerometers are observable and verifiable by others. In addition, professionals reported that many clients tend to overestimate themselves: “*Clients overestimate themselves when it comes to exercise, pedometers don’t. So I give them a pedometer to get a better picture of clients’ behavior*” (P#18, male). In addition, user data patterns and self-reported data from online coaching platforms may provide valuable input for one-on-one sessions or online notifications, for example to address progress, adherence, or barriers. “*I see how often people are online and which assignments are completed and which are not. This gives me important information to help the client and to tailor my treatment*” (P#22, male).

### 3.5. Social Support

Professionals indicate that the provision or arrangement of communication about behavior change, practical help, and/or emotional support from others are important behavioral strategies. A partner, family member, or friend can help clients when a relapse is imminent or food temptations arise. As one participant commented: “*It is so important for clients to share thoughts and to have people around them who can help them*” (P#11, female). Coaches use online platforms that incorporate specific social support functions allowing clients to interact with each other: for example, by sharing experiences and activities with regard to healthy food or physical activity and posting likes and positive comments. These social interactions are, according to professionals, motivating and encouraging. Due to the COVID-19 pandemic, the group sessions in the CLIs were organized online instead of physically. These online meetings were less well received. Interviewees described that the lack of physical proximity hampered the interactivity within the group and perceived more problems with the recognition of emotional expressions. “*Recognizing emotional and physical signals in the group is important. I find it really hard to recognize these signals in the online group sessions*” (P#19, male).

### 3.6. Shaping Knowledge

Professionals provide clients with instructions on how to change a behavior and give clients information about the health consequences of an unhealthy lifestyle to increase awareness, change negative attitudes about a healthy lifestyle, and overcome barriers. Information about food, exercise, physical activity, alcohol consumption, smoking, and sleep is generally provided in group sessions. Individual sessions are used to tailor this information to individual needs and the behavioral stage of change of the client. Shaping knowledge is, especially in the beginning of the trajectory, a key aspect of the existing CLIs. As one participant expressed: “*In the beginning, we mainly gather information about what clients know about nutrition or exercise and what their needs are. Based on this, we give the right health information*” (P#9, male). Experienced eHealth professionals use online information and webinars to provide health information at the right time tailored to treatment goals and clients’ needs. “*We have different online modules. If a client has already the intention to change, he or she gets practical information instead of information about potential health benefits*” (P#33, male).

## 4. Discussion

Changing behavior is evidently challenging, as many people with weight-related health risks struggle to maintain a healthy lifestyle and easy relapse into old habits. This study examined how BCTs are used by health professionals in online- and offline lifestyle interventions to improve physical activity and eating behaviors in clients with weight-related health risks. The results of this study show a diversity of BCTs used in CLIs. Consistent with the results from other research [25,26], the most used BCTs to motivate clients, both in an online and offline setting, were goal setting and action planning, providing feedback and monitoring, facilitating social support, and shaping knowledge. These BCTs were interconnected to help and encourage behavior change in clients with weight-related health risks. Professionals highlighted the importance of goal setting and action planning to prevent setting too cautious or unrealistic exercise and nutrition goals. This should not only be addressed at the start of the intervention but also at various time points during the intervention period. Goal setting is deployed through automatic goal-setting features based on, for instance, accelerometer data or a physical paper diary. Action planning (when, where, and how), goal setting, and the gradual increase in behavioral actions are important to enhance the experience of success [27]. Experience of success, when performing a specific task, provides opportunities to foster self-efficacy which is directly associated with the performance of a healthy behavior [28].

Monitoring was found to be a fundamental BCT in online behavior interventions but less performed in offline behavior interventions. Monitoring to gain information about someone’s behavior takes place through food diary apps, smartwatches, or smartphone-based accelerometers. According to health professionals, clients perceive the collection of objective and continuous parameters as useful and meaningful. Furthermore, the presence of monitoring data provides a starting point for feedback for professionals to address fluctuations in behavior patterns and potential relapses. In line with other studies [25,29], feedback seems of great importance and must be provided just-in-time, individualized, and impunitive based on actual performance data. Professionals indicate that video consultation is a convenient route to provide this in-time feedback, thereby improving the adherence and health outcomes of the long-term lifestyle interventions.

Strong evidence indicates that social support from peers, friends, and family is important for successful weight loss [30] and increased physical activity [31,32]. The positive influence of important others on proximity to day-to-day choices and actions may reduce the risk of unhealthy behaviors and increase treatment adherence [33]. Coaches use different platforms with online communities specifically aimed at increasing social support. These functions encourage clients to interact with each other, talk about their experience, share food and physical activity data, and post likes and positive comments. As a consequence of the COVID-19 pandemic, professionals shifted rapidly toward online activities and systems (e.g., TEAMS and ZOOM) to organize group sessions. However, shifting to online group sessions is challenging, and many coaches were unprepared. Online contact modifies the way of (non-verbal) communication (e.g., eye contact and management of silences) and affects the patient–therapist interaction [34], which requires specific skills. In this study, offline coaches reported difficulties in the recognition of emotional expressions, and the lack of physical proximity hampered the interaction within the group. Lastly, professionals indicated that the provision of health information is crucial to change negative attitudes about exercise and healthy food choices. Shaping knowledge is, especially in the beginning of the trajectory, a key aspect of the existing CLIs.

### 4.1. Implications for Clinical Practice

Online coaching (e.g., video consultations) and eHealth interventions (e.g., online monitoring) are effective strategies to provide specific BCTs [16,17,18]. This is particularly important in long-term CLI interventions, which are distinguished by a behavioral change (year 1) and a maintenance phase (year 2). It is important that online BCTs connect well with the characteristics of these two phases. In the initial phase, knowledge about the health consequences of being overweight and insight into one’s own behavior can be supported through online information (shaping knowledge), online goal setting, self-monitoring, and online feedback. The maintenance phase of the CLI interventions is characterized by less contact and few physical meetings. In this stage, online social support (online community) and just-in-time feedback by a professional are promising to prevent relapse. However, although promising, more research is necessary to investigate when and in which situation strategies are most successful in targeting the behaviors of CLIs. Moreover, it is important to gain more insight into what professionals and clients need to be able to use eHealth in the context of CLIs.

### 4.2. Strengths and Limitations

A strength of this study is that we interviewed a heterogenous group of professionals in terms of gender, age, profession, and eHealth experience. As a consequence, we captured a diversity of perspectives related to both online and offline BCTs. This has enhanced the external validity of our findings. A few limitations should also be taken into consideration. First, we used four different researchers who conducted and analyzed the interviews. This may have influenced the data quality. To reduce the risks, we used pre-specified questions and organized weekly meetings in which the researchers discussed the process and findings in order to become familiar with the interviews and analyses performed by the others. Second, due to the situation of the COVID-19 pandemic, the interviews took place online. This may have hampered the communication between interviewer and interviewee. However, we believe that the online interviews gave professionals the opportunity to take part in this study. This may have led to the high number of conducted interviews in this study.

## 5. Conclusions

The results of this study show a key role for monitoring and feedback in CLIs. Translating this into practice, these findings suggest that it will be useful to use monitoring devices combined with video consultations to provide just-in-time feedback based on the client’s actual performance. In addition, it can be useful to incorporate specific social support functions allowing CLI clients to interact with each other. Lastly, the use of online modules to enhance knowledge about health consequences of unhealthy behavior in clients with weight-related health risks do fit with our results.

## Figures and Tables

**Table 1 ijerph-19-00521-t001:** Participant characteristics.

Characteristics	Value (*n* = 38)
**Gender, *n* (%)**	
Female	18 (47)
Male	20 (53)
Age (years), mean (SD)	38.6 (10.6)
Work experience (years), mean (SD)	6.4 (8)
**Profession, *n* (%)**	
Lifestyle coach	14 (37)
Physical therapist	2 (5)
Exercise therapist	10 (26)
Dietician	4 (11)
Sports professional	3 (8)
CLI developer	5 (13)
**Experience with**	
Offline coaching	14 (37)
Online coaching	14 (37)
Development of (online) CLIs	10 (26)

## Data Availability

The data presented in this study are available on request from the corresponding author. The data are not publicly available due to ethical reasons.

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
