# Peer review of "Online and Offline Behavior Change Techniques to Promote a Healthy Lifestyle: A Qualitative Study"

_ijerph, 2022, doi:10.3390/ijerph19010521_

Round 1

Reviewer 1 Report

In this study the authors deal with an interesting topic related to the eHealth application for combined lifestyle interventions, collecting useful information and applying the appropriate methodology. The paper is generally well written and structured, making it easy for the reader to understand the goal of this research. However, in my opinion, there are several points that require to be addressed and better clarified before further proceeding.

#1) Authors should better explain in the introduction what is eHealth, providing a definition, strengths and limitations related to the CLIs and BCTs, going a bit more further than just using well-known “buzzwords” like: “eHealth interventions provide tailored health information, feedback, goal setting, rewards and reminders”. As well, please define better the concept of e-coaching.

#2) Section 3.2 Behavior Change Techniques, seems a bit overlooked. Would be much beneficial for the readers to offer more information about the BCTs and how is linked with the study goals and results. Moreover, authors reported on (page 4, lines 147-178): “Goal setting is a frequently reported BCT strategy to promote healthy eating and physical activity behaviour”. A reference musty be provided to support such a statement.

#3) Authors said on (page 4 lines 170-181): “Professionals indicate that feedback should be personalized, goal oriented and based on actual performed behavior. However, the feedback provided by health professionals as part of the lifestyle interventions is mostly based on clients’ stories, rather than objective performance data. As one professional put it: “I don’t monitor exercises, fat intake or other health indicators. I do not have the sources to track these things” (P#4, female). Professionals who do use online behavior change interventions point out that activity trackers enable them to provide accurate feedback. Professionals use different applications to monitor food intake (e.g. calory tracker) or physical activity (e.g. fitbit, MyFintessPal, and an application named ‘Ommetje’ [Dutch for ‘Detour’]). These applications are particular promising for collecting objective and continuous physical activity parameters (e.g. steps, minutes, intensity) and have the ability to transfer data to online platforms and mobile phones”. So, is not clear to me if objective data performance is better (and good) to set properly and monitor the client's goals and performance. Please clarify that!

#4) Have the authors noted any (interesting) different approaches across the professionals involved? Perhaps lifestyle coaches and physical therapists had different styles or different goals in following their BCTs strategy. If present, reporting such kinds of differences could be interesting to map the different styles for the different profiles considered.

#5) Social support is (as always) a good indicator for behavioural change and lifestyle influences. However, it seems that online sessions were less useful and perceived as a barrier in recognising emotional and physical signals. That’s an interesting point, which requires more attention and further discussion in the manuscripts. Please discuss a bit more about that.

#6) To provide more practical tools for professionals, perhaps a table that synthesizes the identified BCTs and the related implications and practices that emerged from the interviews notes (quotes) can be added in section 4.1 Implications for clinical practices. 

Reviewer 2 Report

The manuscript "Online and Offline Behavior Change Techniques to Promote a Healthy Lifestyle: A Qualitative Study" presented for evaluation requires development and extension of the presented results.

The topic is very important and up-to-date, but it has been treated superficially. The research methodology, group selection and presentation of the results raise objections.

There are certain parts of the article where the main purpose of the paper becomes unclear and other informations are quite confusing.  It is highly advisable for the authors to clearly state their intentions. I would like to present some detailed suggestions to improve the manuscript below:

[29] Does this phenomenon only occur in Netherlands?

[67] The main aim is well established, but there is no complete nor synthetic answer in the paper if it was actually achieved. I have my concerns?

[133] In my opinion, the study group is poorly selected. The population of 38 people is really small and there is no point in differentiating it according to the form of work. For example, we do not know how many patients have used the online / offline / blended form.

What work experience did they have (years of practice, number of patients, workplace)

What does "blended" mean? Did every patient have filled a online / offline form (half and half) or was some of the patients consulted only online and some only offline?

[140] Where is the data analysis?

[276] The fact that the group is heterogeneous and so small is the weakest element of the work. If it is a representative group, provide statistical evidence that proves it.

[290] In principle, such conclusions can be made without any research. Two groups were not compared, but basically 3 (online, offline / blended), it could be very interesting and innovative. In the current version, the results are either not innovative, or have not been properly developed and presented.

Reviewer 3 Report

Although integrating eHealth into existing CLIs is an interesting topic, I am a bit disappointed with the research content. I suggest that the author should strengthen the data analysis method. The current results are really too rough.

Round 2

Reviewer 2 Report

N/A